# Metabolomics-Based Machine Learning for Predicting Mortality: Unveiling Multisystem Impacts on Health

**DOI:** 10.3390/ijms252111636

**Published:** 2024-10-30

**Authors:** Anniina Oravilahti, Jagadish Vangipurapu, Markku Laakso, Lilian Fernandes Silva

**Affiliations:** 1Institute of Clinical Medicine, Internal Medicine, University of Eastern Finland, 70210 Kuopio, Finland; anniina.oravilahti@uef.fi (A.O.); jagadish.vangipurapu@uef.fi (J.V.); markku.laakso@uef.fi (M.L.); 2Department of Medicine, Kuopio University Hospital, 70200 Kuopio, Finland; 3Department of Medicine, Division of Cardiology, David Geffen School of Medicine, University of California, Los Angeles, CA 90095, USA

**Keywords:** mortality, metabolites, metabolism, artificial intelligence, aging

## Abstract

Reliable predictors of long-term all-cause mortality are needed for middle-aged and older populations. Previous metabolomics mortality studies have limitations: a low number of participants and metabolites measured, measurements mainly using nuclear magnetic spectroscopy, and the use only of conventional statistical methods. To overcome these challenges, we applied liquid chromatography–tandem mass spectrometry and measured >1000 metabolites in the METSIM study including 10,197 men. We applied the machine learning approach together with conventional statistical methods to identify metabolites associated with all-cause mortality. The three independent machine learning methods (logistic regression, XGBoost, and Welch’s *t*-test) identified 32 metabolites having the most impactful associations with all-cause mortality (25 increasing and 7 decreasing the risk). From these metabolites, 20 were novel and encompassed various metabolic pathways, impacting the cardiovascular, renal, respiratory, endocrine, and central nervous systems. In the Cox regression analyses (hazard ratios and their 95% confidence intervals), clinical and laboratory risk factors increased the risk of all-cause mortality by 1.76 (1.60–1.94), the 25 metabolites by 1.89 (1.68–2.12), and clinical and laboratory risk factors combined with the 25 metabolites by 2.00 (1.81–2.22). In our study, the main causes of death were cancers (28%) and cardiovascular diseases (25%). We did not identify any metabolites associated with cancer but found 13 metabolites associated with an increased risk of cardiovascular diseases. Our study reports several novel metabolites associated with an increased risk of mortality and shows that these 25 metabolites improved the prediction of all-cause mortality beyond and above clinical and laboratory measurements.

## 1. Introduction

Identifying predictors for all-cause mortality is essential to improve the risk assessment in medical decision-making and elucidate the pathways leading to disease outcomes. Studies with detailed longitudinal clinical data surrounding death give the opportunity to better understand the risk factors of mortality. Metabolic biomarkers for all-cause mortality reflect multimorbidity among middle-aged and older people, and not only for specific diseases [1]. However, our understanding of metabolic changes underlying mortality and aging remains incomplete.

Previous studies on all-cause mortality have focused mainly on clinical and laboratory measurements or the identification of metabolic biomarkers for specific diseases and conditions, including cardiovascular diseases, type 2 diabetes, and chronic kidney disease [2,3,4,5,6]. Three previous studies identified metabolic biomarkers for all-cause mortality by applying nuclear magnetic resonance (NMR) spectroscopy. The strength of these studies lies in their large sample sizes, which allows the replication of findings in other cohorts. The limitation of these studies is that the number of metabolites measured was low, from 98 to 226 [7,8,9]. The sensitivity of NMR is low compared to the liquid chromatography mass spectrometry (LC-MS/MS) method. The LC-MS/MS method detects a large pool of metabolites (>1000), and therefore it plays a dominant role in the metabolomics field. Mass spectrometry is intrinsically a highly sensitive method for the detection, quantitation, and structure elucidation of metabolites [10]. Wang et al. were the first to apply the LC/MS approach to investigate the association of 243 metabolites with mortality in 13,512 individuals, and found that high levels of N2, N2-dimethylguanosine, pseudo uridine, N4-acetylcytidine, 4-acetamidobutanoic acid, N1-acetylspermidine, and lipids with fewer double bonds were associated with an increased risk of all-cause mortality [3].

Previous studies trying to find metabolic biomarkers for mortality have applied conventional statistics, which have limitations due to high internal correlations, class diversity, and exposure–outcome disparities. Artificial intelligence includes several technologies of the machine learning (ML) approach and, therefore, it is well suited to mortality studies. It focuses on the empirical prediction of an outcome in contrast to traditional statistical methods [11]. Several methods, including ML tools, have been applied to metabolomics to create clinical prediction models. ML methods can analyze thousands of predictors effectively by optimizing predictive performance while capturing complicated patterns in the data, including non-linear relationships. It is especially well suited to studies applying metabolomics in mortality data, as the mechanisms of action and interactions between the metabolites are biologically diverse and interconnected [11].

Previously published studies have several limitations, especially the low number of participants and metabolites measured, the lack of modern statistical methods to analyze the data, and innovations and contributions to generate risk models for clinical practice. We hypothesized that identifying metabolites by the LC-MS/MS platform and applying parallel, conventional statistical methods with ML tools can improve the identification of metabolites associated with all-cause mortality. This approach also gives us tools to generate risk scores to identify people at high risk of mortality. Our study is the first to apply the LC-MS/MS metabolomics-based method together with ML tools to investigate metabolites associated with all-cause mortality in a large population-based cohort including 10,197 Finnish men.

## 2. Results

### 2.1. Baseline Characteristics of the Study Population

Table 1 shows the baseline characteristics of the participants of the METSIM study (METabolic Syndrome In Men) who are alive (*n* = 8851) and the participants who died during the follow-up (*n* = 1346). Compared to the living participants, the participants who died during the follow-up were older, had higher body mass index (BMI) and waist circumference, were more often smokers, had higher systolic blood pressure, lower low-density lipoprotein cholesterol (LDLC) levels, higher total triglycerides, higher fasting glucose levels, higher high-sensitivity C-reactive protein (hs-CRP) levels, higher rates of type 2 diabetes (T2D), higher creatinine levels, higher urinary albumin excretion (UAE) rates, and lower estimated glomerular filtration rates (eGFRs). No difference between the two groups was observed for alanine transferase (ALT).

### 2.2. Most Impactful Metabolite Predictors of Mortality Identified by ML Tools

Appendix A shows the post-processing of the data from the original dataset (*n* = 1540) to the identification of the most impactful metabolites predicting mortality based on each of the three ML methods: Welch’s *t*-test, XGBoost (eXtreme Gradient Boosting), and Logistic Regression. We obtained the final set of 32 metabolites shared by all three ML models. Appendix A shows the relative importance of the metabolites on mortality prediction by absolute SHAP (SHapley Additive exPlanations) values. The SVM (support vector machine) model for binary classification of mortality prediction yielded the following performance metrics: precision 0.87, accuracy 0.84, and ROC-AUC 0.75 (area under the curve of the receiver operating characteristic curve). The ROC-AUC values were evaluated from an independent test set. In the logistic regression model of the 32 metabolites, the performance of the corresponding metrics was as follows: precision 0.81, accuracy 0.85, and ROC-AUC 0.77, and for the XGBoost binary classifier model, precision 0.65, accuracy 0.72, and ROC-AUC 0.79. Figure 1 shows the ROC-AUC curves for the three models. The results were very similar across the three models.

Figure 2 shows a SHAP summary plot of the 32 most impactful metabolite predictors of mortality at the population level. Positive SHAP values indicate an increased risk for mortality and negative SHAP values, a protective effect. Each dot corresponds to a single observation. Increased metabolites are shown in red and decreased metabolites in blue. For example, SHAP values indicate that N-acetylcarnosine decreases the risk of mortality equally in the study population. The 5-(galactosylhydroxy)-L-lysine prediction pattern plot has a long tail, where low levels indicate an increased risk of mortality, whereas increased levels indicate a similar increased risk of mortality in the entire population. The SHAP summary plot gives an explanative pattern for the prediction for each metabolite.

We applied ML methods, logistic regression, Welch’s *t*-test, and XGBoost, to identify the most impactful metabolites for short-, intermediate-, and long-term mortality (Appendix A). Short-term mortality included 185 mortality cases with a follow-up time of 2.41 ± 1.35 years, intermediate-term mortality included 495 cases with a follow-up time of 6.09 ± 1.98 years, and long-term mortality included 666 cases with a follow-up time of 11.04 ± 1.97 years.

In the short-term mortality group, the three most impactful metabolites were 9-hydroxystearate, N1-methyladenosine, and lactate. In the intermediate-term mortality group, the three most impactful metabolites were 3-ureidopropionate, o-cresol sulfate, and C-glycosyltryptophan. In the long-term mortality group, the three most impactful metabolites were behenoyl dihydrosphingomyelin (d18:0/22:0), dehydroepiandrosterone sulfate (DHEA-S), and malate. The Veen diagram (Appendix A) shows that none of the metabolites were shared between the short-, intermediate-, and long-term mortality groups. The short- and intermediate-term mortality groups shared three metabolites, intermediate- and long-term mortality shared five metabolites, and short- and long-term mortality shared three metabolites.

### 2.3. Clustered Heatmap of the 32 Metabolites

The heatmap shows the correlation of the 32 most impactful metabolites in mortality prediction (Appendix A). The five most significant metabolites predicting mortality were 2-hydroxyfluorene sulfate, N-acetylcarnosine, pregnenetriol sulfate, lignoceroyl sphingomyelin, and 5-(galactosyl-hydroxy)-L-lysine. Hierarchical clustering shows that metabolites having similar biological functions cluster together, especially metabolites belonging to the amino acid pathway or sphingomyelins. Metabolites that increase the risk of mortality cluster together, suggesting similar prediction patterns for mortality. Correspondingly, metabolites decreasing the risk of mortality clustered together and had an inverse correlation with the metabolites increasing the risk of mortality.

### 2.4. Cox Regression Analysis of Metabolites Associated with Mortality Risk

We performed a Cox proportional hazards regression analysis for the metabolites identified by the machine learning models (Table 2). The analysis was used to estimate the hazard ratios (HRs) and 95% confidence intervals (CIs) for each metabolite with all-cause mortality. The Cox regression model was adjusted for age. Metabolite concentrations were standardized prior to analysis.

We conducted Cox regression analyses to assess the effects of the clinical and laboratory measurements and the 25 metabolites on the risk of mortality, both individually and in combination. Clinical and laboratory measurements including age, BMI, waist circumference, smoking, systolic blood pressure, LDLC, total triglycerides, fasting glucose, hs-CRP, creatinine, T2D, and UAE significantly increased the risk of mortality (HR 1.76, 1.60–1.94, *p* = 1.7 × 10^−29^). The 25 metabolites alone also increased the risk of mortality significantly (HR 1.89, 1.68–2.12, *p* = 5.2 × 10^−27^). In the model including both the clinical and laboratory measurements and the 25 metabolites, the risk of mortality further increased (HR 2.00, 1.81–2.22, *p* = 3.4 × 10^−42^), suggesting that the 25 metabolites increased the risk of mortality beyond the clinical and laboratory risk factors for mortality.

## 3. Discussion

Previous studies of all-cause mortality applying the metabolomics approach have been heterogeneous in the size of the studies, the number of metabolites included in the studies, the platforms to measure metabolites, and the statistical methods. We applied ML tools (SVM, XGBoost, and logistic regression) to identify the most impactful metabolites associated with mortality, and identified 32 metabolites, 25 metabolites increasing and 7 metabolites decreasing the risk of mortality. Twenty of these metabolites were novel, covering several metabolic pathways, lipids, amino acids, carbohydrates, xenobiotics, energy metabolism, nucleotides, endocannabinoids, and peptides. These metabolites are known to be associated with damage in the key human body systems, including the cardiovascular, renal, respiratory, endocrine, and central nervous systems (Figure 3).

When we compared our findings with the previous two large studies, we found that only one metabolite, histidine, was previously reported to be associated with decreased mortality in the study of Deelen et al. [9], and another metabolite, 3-ureidopropionate, was associated with increased mortality in the study of Wang et al. [4]. The number of metabolites measured varied significantly across these studies. Our study included >1000 metabolites whereas the Deelen et al. study [9] included 226 metabolites and Wang et al.’s study [4], 243 metabolites.

In our study, seven of the metabolites damaged multiple body systems, including three novel metabolites (3-amino-2-piperidone, C-glycosyltryptophan, and 5-(galactosyl)-L-lysine), and four previously reported metabolites (N-acetylphenylalanine, homocitrulline, homoarginine, and 5-hydroxyhexanoate) [12,13,14,15]. Disruptions in the ornithine cycle result in an increased abundance of 3-amino-2-piperidone (Appendix A), resulting in enhanced coagulation [16]. Hypercoagulation increases the risk of myocardial infarction and stroke, pulmonary embolism, pulmonary infarction, and renal thrombosis [17].

N-acetylphenylalanine and C-glycosyltryptophan have been associated with albuminuria [18] and cardiovascular mortality. C-glycosyltryptophan accelerates peripheral artery disease in patients with type 2 diabetes and is associated with a decrease in kidney function, pulmonary hypertension, and impaired lung function [19,20]. Increased concentrations of 5-(galactosylhydroxyl)-L-lysine, a glycosylation product of hydroxylysine (Appendix A), have been found in patients with pulmonary artery hypertension and in patients with impaired kidney function [20,21].

Homocitrulline, a carbamylation product, has been reported to be associated with morbidity and mortality from chronic heart failure, coronary artery disease, and chronic kidney disease [22,23]. Cyanate-induced carbamylation generates homocitrulline from lysine (Appendix A). Elevated cyanate concentrations related to impaired kidney function and inflammation increase homocitrulline concentration [24]. Carbamylation prevents LDLC binding to its receptor, resulting in cholesterol accumulation, macrophage foam-cell formation, and an increased risk of coronary artery disease [25].

**Figure 3 ijms-25-11636-f003:**
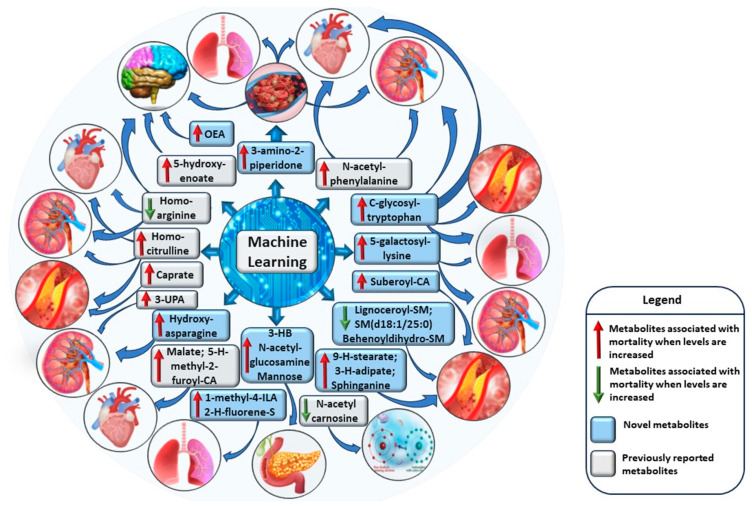
The impact of machine learning-identified metabolites on multiple body systems. Metabolites damaging cardiovascular system when levels are decreased: lignoceroylsphingomyelin, sphingomyelin (d18:1/25:0), behenoyldihydrosphingomyelin, and homoarginine. Metabolites damaging renal system when levels are increased: 3-amino-2-piperidone, N-acetylphenylalanine, C-glycosyltryptophan, 5-galactosyllysine, hydroxyasparagine, 3-ureidopropionate, and homocitrulline. Metabolites damaging renal system when levels are decreased: homoarginine. Metabolites damaging respiratory system when levels are increased: 3-amino-2-piperidone, C-glycosyltryptophan, 5-galactosyllysine, 1-methyl-4-imidazoleacetate, and 2-hydroxyfluorene sulfate. Metabolites damaging central nervous system when levels are increased: 3-amino-2-piperidone, oleoylethanolamide, and 5-hydroxyhexanoate. Metabolites damaging endocrine system when levels are increased: (S)/(R)-hydroxybutyrate, N-acetylglucosamine, and mannose. Metabolites damaging antioxidant system when levels are decreased: N-acetylcarnosine. Detailed information about mechanisms exerted by these metabolites can be found in Appendix A. Abbreviations: CA, carnitine; H, hydroxy; HM, hydroxybutyrate; ILA, imidazoleacetate; OEA, oleylethanolamide; SM, sphingomyelin; UPA, ureidopropionate. Metabolites damaging cardiovascular system when levels are increased: 3-amino-2-piperidone, N-acetylphenylalanine, C-glycosyltryptophan, subeoylcarnitine, 9-hydroxystearate, 3-hydroxyadipate, sphinganine, malate, 5-hydroxymethyl-2-furoylcarnitine, caprate, and homocitrulline. Lysine can replace ornithine in the urea cycle and combine with arginine to form homoarginine (Appendix A). An increase in homoarginine was inversely associated with mortality in our study, in agreement with the findings in the LURIC and 4D studies [26]. Homoarginine acts as a nitric oxide precursor, enhancing endothelial function [26]. Elevated homocitrulline and decreased homoarginine result in disruption of the lysine pathway and increases the risk of mortality [15].

We found 22 metabolites known to impair specific body systems, 7 novel metabolites contributing to coronary artery disease (9-hydroxystearate, 3-hydroxyadipate, sphinganine, lignoceroyl-SM, SM (d18:1/25:0), behenoyl dihydro-SM, and suberoylcarnitine), and 1 previously reported metabolite, caprate [15] (Appendix A). Hydrofluoroalkanes, 9-hydroxystearate, and 3-hydroxyadipate can be incorporated into chylomicrons, which contribute to an increase in very low-density lipoprotein particles. Additionally, oxidized LDLC plays an important role in atherosclerosis by inducing monocyte chemotactic protein 1 and scavenger receptors [27], resulting in pro-inflammatory mechanisms.

Sphinganine, a ceramide precursor (Appendix A), inhibits LDLC esterification and contributes to the accumulation of free cholesterol in perinuclear vesicles resulting in cellular toxicity and death [28]. Cholesterol accumulation releases proteases, cytokines, and prothrombotic molecules, contributing to plaque instability, rupture, and vascular occlusion [29]. Three sphingomyelins (lignoceroyl-sphingomyeline, sphingomyeline (d18:1/25:0), and behenoyl dihydro-sphingomyeline) were associated with decreased all-cause mortality in our study. Sphingomyelins are crucial for cell membrane structure and they prevent the deleterious effects of ceramides on endothelial dysfunction, cell apoptosis, and atherosclerosis [30].

Suberoylcarnitine, a medium-chain dicarboxylic acylcarnitine, increases the risk of coronary artery disease attributable to altered mitochondrial fatty acid oxidation and omega-oxidation [31]. Caprate, a saturated fatty acid, has been reported to be associated with increased mortality [32]. Saturated fatty acids increase coagulation, inflammation, insulin resistance, and the risk of type 2 diabetes, cardiovascular diseases, cancer, frailty, and all-cause mortality [33].

We found two metabolites linked to the cardiovascular system, one novel association with 5-hydroxymethyl-2-furoylcarnitine and one previously reported association with malate [12] (Appendix A). 5-hydroxymethyl-2-furoylcarnitine, a dietary component, has been associated with ischemic heart disease [34]. Two metabolites in our study impair the renal system (Appendix A), one novel association with hydroxyasparagine and one previously reported association with 3-ureidopropionate (3-UPA) [22]. 3-UPA (Appendix A) increases mortality independently of kidney disease in patients with liver cirrhosis [35].

We confirmed that N-acetylcarnosine and histidine decreased the risk of mortality [26,36]. N-acetylcarnosine and histidine are carnosine metabolites (Appendix A) known for their antioxidative properties [37]. These metabolites effectively inhibit glucose-induced oxidation and glycation in human LDL, countering aging-related changes in protein oxidation, glycation, and advanced glycation end-product (AGE) formation [38].

We discovered two novel metabolites linked to respiratory system damage, 1-methyl-4-imidazoleacetate and 2-hydroxyfluorene sulfate (Appendix A). 1-methyl-4-imidazoleacetate is the main histamine metabolite (Appendix A) and increases significantly during asthma attacks [39]. Tobacco smoking increases the concentration of 2-hydroxyfluorene sulfate, which is a potent carcinogen in tobacco [40]. We identified a novel metabolite oleoylethanolamide, an important metabolite impacting the central nervous system (Appendix A). Oleoylethanolamide induces anorexia by stimulating vagal sensory nerves and activating PPAR-alpha [41]. Anorexia is associated with an elevated risk of all-cause mortality [42].

We found three novel metabolites impacting the endocrine system, S- and R-3-hydroxybutyrylcarnitine (S-3HB and R-3HB) and mannose (Appendix A), confirming previously the reported association with N-acetylglucosamine [43]. R-3HB-carnitine contributes to insulin resistance in mice and can cause hypoketotic-hypoglycemia, metabolic acidosis, hyperammonemia, and fatty liver disease [44]. Mannose glycates proteins and enhances the formation of AGEs in several diseases, including diabetic nephropathy, atherosclerosis, and neurodegenerative diseases [45]. N-acetylglucosamine/N-acetylgalactosamine generates GlycA, which is associated with cardiovascular diseases and diabetes [46].

We found that the metabolite signatures regulating short-term, intermediate-term, and long-term mortality were very different. Only three metabolites were shared between short-term and long-term mortality. Metabolites associated with short-term mortality reflect acute stress and energy metabolism. N1-methyladenosine is required for RNA methylation and rapid cellular stress adaptation [47].

Lactate and succinate are involved in acute stress responses and fast metabolic energy [48]. Succinate, a key metabolite in the Krebs cycle, activates hypoxia signaling [49] whereas the metabolites associated with long-term mortality, such as dehydroepiandrosterone sulfate (DHEA-S) and beta-cryptoxanthin, regulate chronic inflammation and oxidative stress. A decrease in DHEA-S concentration increases inflammation and has an impact on long-term health [50]. Beta-cryptoxanthin has antioxidant effects and is protective against oxidative stress [51].

The main causes of death in our study were cancers (28%) and cardiovascular diseases (25%). Interestingly, we did not find any metabolite associated with the risk of cancer but instead, 13 metabolites were associated with cardiovascular diseases (myocardial infarction, coronary artery disease, heart failure, and pulmonary artery hypertension). This gives an excellent possibility to use these metabolites as markers for the risk of cardiovascular diseases.

In summary, ML successfully identified a precise set of metabolites associated with an increased risk of all-cause mortality, emphasizing the significant role of metabolism in aging and different diseases. Most of the 32 metabolites we discovered were novel and regulated coagulation, cytokine release, lipid oxidation, inflammation, cellular toxicity, insulin resistance, urea and malate–aspartate cycle dysregulation, and especially the risk of cardiovascular diseases. Several of these metabolites can simultaneously harm multiple body systems (Appendix A). These metabolites offer a more accurate representation of general health compared to traditional clinical parameters and laboratory measurements.

## 4. Materials and Methods

### 4.1. Study Population

Our study population, the METSIM study, is a randomly selected population-based cohort comprising 10,197 men, aged from 45 to 73 years at baseline, and recruited from Kuopio and the surrounding communities in Eastern Finland [52]. A total of 7090 individuals participated in a 12-year follow-up study. The mean age of death in the participants was 76.0 ± 6.7 years (mean ± standard deviation, SD). The main causes of death were cancers (28%), cardiovascular diseases (25%), and neurological diseases (11%). The METSIM study was approved by the Ethics Committee of the University of Eastern Finland and Kuopio University Hospital and was conducted in accordance with the Declaration of Helsinki. All participants gave written informed consent.

### 4.2. Clinical and Laboratory Measurements

BMI was calculated as weight in kilograms divided by height in meters squared. Waist circumference was measured to the nearest 0.5 cm. LDLC and total triglycerides were measured by enzymatic colorimetric tests (Konelab System Reagents). Plasma glucose was measured by enzymatic hexokinase photometric assay (Konelab Systems reagents; Thermo Fischer Scientific, Vantaa, Finland). hs-CRP was determined by an Immulite 2000 High Sensitivity CRP assay (Diagnostic Products Corp., Los Angeles, CA, USA). Creatinine was determined by the Jaffe method. ALT was assessed by enzymatic photometric test. UAE rate was determined by the Immunoturbidimetric method (Konelab Albumin/Microalbuminuria system reagents, REF no 981660, Thermo Electron Corp, Vantaa, Finland) from the first urine sample in the morning (µg/minute). The eGFR was calculated with the Cockroft–Gault formula [52].

### 4.3. Metabolomics

Non-targeted metabolomics profiling was performed at Metabolon, Inc. (Morrisville, NC, USA) on EDTA plasma samples obtained after overnight fasting from 10,188 participants at baseline, as previously described in detail [53]. The Metabolon DiscoveryHD4 platform was applied to identify the metabolites. All samples were processed together for peak quantification and data scaling. We quantified raw mass spectrometry peaks for each metabolite using the area under the curve and evaluated the overall process variability by the median relative standard deviation for the endogenous metabolites present in all 20 technical replicates in each batch. We adjusted for variation caused by day-to-day instrument tuning differences and columns used for biochemical extraction by scaling the raw peak quantifications to the median for each metabolite by the Metabolon batch. Instrument variability was assessed by calculating the median relative standard deviation (RSD) for internal standards added to each sample before injection into the mass spectrometers. The acceptance criterion for instrument variability was a median RSD of 5% or lower, which was obtained in our study. Overall process variability was determined by calculating the median RSD for all endogenous metabolites in technical replicates, with an acceptance criterion of a median RSD of 15% or lower. Our study achieved a median RSD of 8% which meets Metabolon’s acceptance criteria ensuring high data quality.

### 4.4. Machine Learning

We included 1540 metabolites in our study (Appendix A). We filtered out 596 metabolites, of which 416 metabolites had more than 50% of missing values, and 180 metabolites had no identification available. We included 945 normalized metabolites in statistical analyses. Missing values were set to NaN to utilize the XGBoost’s built-in function for handling missingness. We addressed the class imbalances in mortality events by reducing the size of the dataset from 10,000 to 6683 to ensure that our machine learning models perform effectively. The final dataset consisted of 945 metabolites as variables and 6683 samples as datapoints (Appendix A).

We applied three distinct methods, logistic regression, Welch *t*-test, and XGBoost, to the entire preprocessed dataset to perform feature selection and rank the most significant metabolites predicting mortality (Appendix A). This approach has previously been used to identify the most impactful metabolites associated with a disease or condition [54]. We performed Welch’s *t*-test for each metabolite to determine how well it discriminates between the individuals who died during the follow-up period. The 200 most discriminating metabolites with the lowest q-value according to the Welch *t*-test were selected. We examined all metabolites individually with logistic regression for their discrimination ability. Metabolites were ranked based on the magnitude of the ROC-AUC curve (area under the receiver operating characteristic curve) with logistic regression, and the top 200 metabolites were selected. We performed XGBoost tree binary classification for the entire dataset of 945 metabolites (Appendix A). We sorted the metabolites in the order of magnitude according to their importance value produced by the XGboost model. A total of 154 metabolites were selected based on a SHAP feature importance value greater than 0.012. The final set of the most impactful 32 metabolites was selected from the intersection of the top-ranked metabolites identified by these three methods.

We built the three prediction models to evaluate the predictive power of 32 selected metabolites for all-cause mortality: the support vector machine (SVM) model, logistic regression model, and XGBoost binary classifier model. The feature selection process, implementation, and evaluation of the ML models were performed by Python 3.8.10 version. The Python XGBoost function (version 1.3.3) XGBClassifier and SHAP version (0.38.1) were used to build the explainable ML model of mortality. XGBoost is an implementation of the gradient-boosted decision tree, and the algorithm is designed for speed and performance. Shapley Additive exPlanation (SHAP) values are based on classic game-theoretic Shapley values [55] which are used to explain predictions generated by machine learning models [55]. The final set of hyperparameters used in the XGBoost mortality prediction model is presented in Appendix A.

We applied hyperparameter tuning to regulate the overfitting caused by complex tree-based algorithms with numerous variables. Model complexity was reduced by restricting maximum tree depth (max depth) and increasing the minimum sum of instance weights (min_child_weight), which both lead to a more conservative model. The randomness of the model was evaluated by the colsample bytree parameter which restricts the number of variables used in one tree to make training more robust. We used the Python seaborn clustermap function with the clustering method Nearest Point Algorithm “single” and Euclidean distance to perform the hierarchical clustering algorithm.

### 4.5. Statistical Analyses

We conducted statistical analyses using IBM SPSS Statistics, version 25. We log-transformed all continuous variables having skewed distribution. We applied the Cox regression analysis to associate the metabolites with all-cause mortality and presented the results as hazard ratios (HRs) and their 95% confidence intervals (CIs). When analyzing the 25 metabolites, the predictors of the mortality scores were derived by adding metabolites weighted by their regression coefficients. We tested the Cox proportionality assumption for the metabolites using the survival and survminer packages in R and found that a fitted Cox regression model adequately described the data. *p* < 4.06 × 10^−5^ (Bonferroni correction for 1.232 metabolites) was considered statistically significant. We used one-way ANOVA and Chi-square tests to assess the differences in clinical traits and metabolites between the cases (deceased) and the controls (alive).

## 5. Conclusions

Our study has several strengths, including a large METSIM cohort, a validated metabolomics platform including >1000 metabolites, several novel findings, and robust data analysis. Our study applied ML methods to identify the metabolites associated with all-cause mortality. Most of the metabolites were novel and regulate coagulation, lipid oxidation, endothelial dysfunction, and inflammation, highlighting the role of metabolic changes related to aging and different diseases, particularly to cardiovascular complications. Our study shows that metabolomics studies need to include a high number of participants and metabolite measures to identify novel metabolites and metabolic pathways. Our findings offer valuable insights into metabolic pathways and potential biomarkers for future research.

Our study has implications for clinical practice. Using Cox regression analyses, we were able to compare the effects of clinical and laboratory measurements and the 25 most impactful metabolites and their combination on the risk of all-cause mortality. We found that clinical and laboratory measurements increased the risk of mortality by 1.76-fold, the 25 metabolites by 1.89-fold, and the combination of these two by 2.00-fold. Our study shows that the metabolites increasing the risk of all-cause mortality significantly improves the prediction of mortality beyond and above clinical and laboratory measurements. Most of the novel metabolites were associated with an increased risk of cardiovascular diseases. Therefore, our method to calculate a risk score by combining metabolites and clinical and laboratory measurements is especially suited to identify patients with a high risk of cardiovascular diseases. The limitations of our study are that it included only middle-aged Finnish men, and therefore our results need to be confirmed in females and non-Finnish populations.

## Figures and Tables

**Figure 1 ijms-25-11636-f001:**
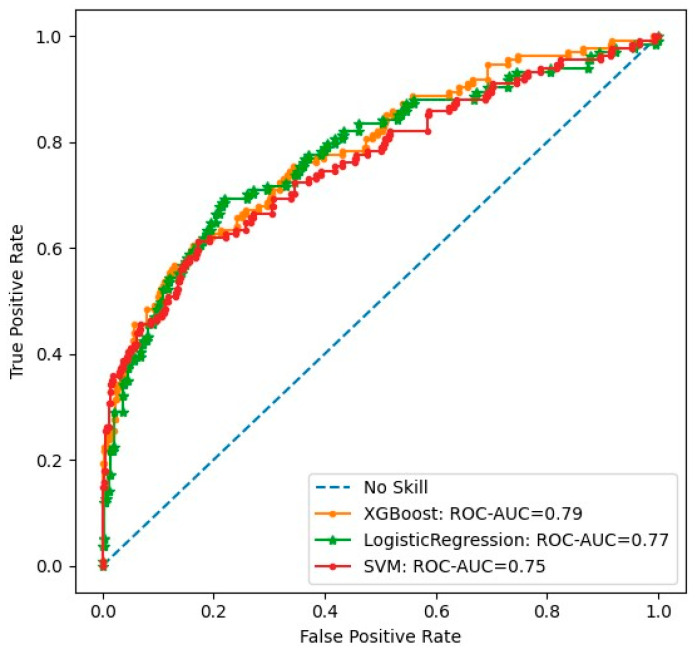
ROC-AUC curves for XGBoost, Logistic Regression, and SVM models.

**Figure 2 ijms-25-11636-f002:**
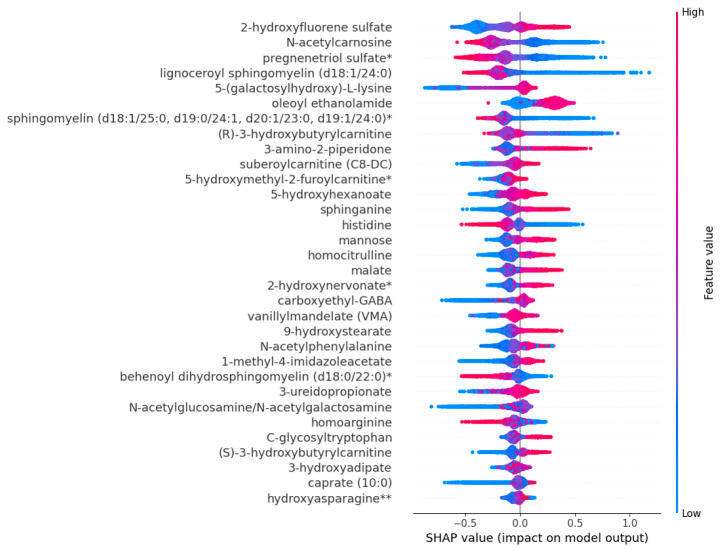
SHAP summary plot of the 32 most impactful predictors of mortality. A positive SHAP value means an increased risk prediction on mortality and a negative SHAP value indicates a protective effect. Each dot corresponds to a single observation and higher values of the variable are shown in red and lower values in blue. * indicates a tentatively identified metabolite; ** signify a well-characterized compound with minor identification uncertainty.

**Table 1 ijms-25-11636-t001:** Baseline characteristics of the participants of the METSIM study.

	Alive		Deceased		
Variable	*n*	Mean ± SD	*n*	Mean ± SD	*p*
Age (years)	8851	56.9 ± 6.9	1346	62.5 ± 6.5	2.0 × 10^−161^
Body mass index (kg/m^2^)	8849	27.2 ± 4.0	1344	28.2 ± 5.0	3.1 × 10^−15^
Waist (cm)	8848	98.2 ± 11.1	1343	102.3 ± 11.5	3.4 × 10^−31^
Smoking (%) *	8851	16.8	1344	26.4	1.5 × 10^−16^
Systolic blood pressure (mmHg)	8851	137.5 ± 16.4	1345	143.3 ± 18.0	7.0 × 10^−31^
Type 2 diabetes (%) *	8851	11.9	1345	27.2	1.3 × 10^−44^
LDLC (mmol/L)	8847	3.34 ± 0.89	1346	3.10 ± 0.92	2.0 × 10^−19^
Triglycerides (mmol/L)	8850	1.45 ± 0.97	1346	1.58 ± 1.23	4.8 × 10^−07^
Fasting glucose (mmol/L)	8851	5.92 ± 0.98	1346	6.30 ± 1.70	6.5 × 10^−27^
hS-CRP (mg/L)	8850	2.01 ± 4.27)	1345	3.44 ± 5.62	2.5 × 10^−43^
Creatinine (umol/L)	8851	83.5 ± 13.4	1346	86.7 ± 24.5	5.1 × 10^−8^
Urinary albumin excretion rate (ug/min)	8740	17.7 ± 95.5	1311	69.1 ± 31.9	1.0 × 10^−33^
eGFR (mL/min/1.73 m^2^)	8850	88.7 ± 12.1	1345	83.4 ± 14.9	6.4 × 10^−52^
ALT (U/L)	8851	32.5 ± 21.2	1346	32.1 ± 22.0	0.562

* Chi-square test.

**Table 2 ijms-25-11636-t002:** Cox regression analysis of metabolites associated with the risk of mortality.

HMDB	Metabolite	Cases	Total	HR (95% CI)	*p*	Novel
Amino Acids						
HMDB0341329	Hydroxyasparagine	1345	10,169	1.23 (1.16–1.29)	2.1 × 10^−15^	Yes
HMDB0000177	Histidine	1345	10,188	0.85 (0.81–0.88)	3.2 × 10^−15^	No
HMDB0000670	Homoarginine	1345	10,188	0.87 (0.82–0.91)	1.8 × 10^−8^	No
HMDB0002820	1-methyl-4-imidazoleacetate	1333	10,125	1.25 (1.19–1.29)	<1.0 × 10^−20^	Yes
HMDB0000600	5-(galactosylhydroxy)-L-lysine	1165	8180	1.17 (1.10–1.24)	3.8 × 10^−7^	Yes
HMDB0000512	N-acetylphenylalanine	1317	9959	1.26 (1.19–1.32)	3.0 × 10^−17^	No
HMDB0240296	C-glycosyltryptophan	1345	10,188	1.26 (1.20–1.33)	<1.0 × 10^−20^	Yes
HMDB0000679	Homocitrulline	1304	9837	1.19 (1.13–1.26)	5.5 × 10^−11^	No
HMDB0000323	3-amino-2-piperidone	1344	10,180	1.15 (1.10–1.21)	4.4 × 10^−9^	Yes
HMDB0002201	Carboxyehtyl-GABA	1308	9898	1.14 (1.08–1.21)	4.2 × 10^−6^	Yes
Peptide						
HMDB0012881	N-acetylcarnosine	1340	10,162	0.87 (0.83–0.92)	2.3 × 10^−7^	No
Nucleotides						
HMDB0000026	3-ureidopropionate	1236	9104	1.27 (1.12–1.33)	<1.0 × 10^−20^	No
Fatty acids						
HMDB0000345	3-hydroxyadipate	1054	7933	1.25 (1.18–1.18)	1.1 × 10^−13^	Yes
HMDB0061661	9-hydroxystearate	1191	9011	1.37 (1.30–1.44)	<1.0 × 10^−20^	Yes
-	2-hydroxynervonate	1317	9773	1.37 (1.28–1.46)	<1.0 × 10^−20^	Yes
HMDB0000409	5-hydroxyhexanoate	1125	7220	1.23 (1.16–1.31)	3.4 × 10^−12^	No
HMDB0000511	Caprate (10:0)	1345	10,188	1.22 (1.16–1.28)	1.3 × 10^−14^	No
Sphingolipids						
HMDB0000269	Sphinganine	1257	8796	1.22 (1.15–1.29)	1.7 × 10^−11^	Yes
HMDB0011697	Lignoceroyl sphingomyelin	1136	7896	0.88 (0.31–0.93)	9.9 × 10^−6^	Yes
HMDB0240671	Sphingomyelin (d18:1/25:0)	1136	7893	0.85 (0.80–0.90)	6.8 × 10^−9^	Yes
HMDB0012091	Behenoyl dihydrosphingomyelin	1337	10,008	0.89 (0.51–0.94)	1.1 × 10^−5^	Yes
Acylcarnitines						
-	Suberoylcarnitine (C8-DC)	1163	8684	1.31 (1.24–1.39)	<1.0 × 10^−20^	No
HMDB0013127	(R)-3-hydroxybutyrylcarnitine	1292	9620	1.22 (1.15–1.38)	8.9 × 10^−13^	Yes
-	(S)-3-hydroxybutyrylcarnitine	1334	10,014	1.20 (1.14–1.26)	1.0 × 10^−11^	Yes
Steroids						
-	Pregnenetriol sulfate	1345	10,187	0.89 (0.85–0.94)	9.8 × 10^−6^	Yes
Carbohydrates						
HMDB0000212 HMDB0000215	N-acetylglucosamine/N N-acetylgalactosamine	1334	10,053	1.30 (1.23–1.38)	<1.0 × 10^−20^	No
HMDB0000169	Mannose	1345	10,185	1.22 (1.16–1.29)	9.9 × 10^−13^	Yes
Energy						
HMDB0031518	Malate	1345	10,188	1.33 (1.26–1.39)	<1.0 × 10^−20^	No
Endocannab.						
HMDB0002088	Oleoylethanolamide	1109	7189	1.18 (1.11–1.26)	7.2 × 10^−8^	Yes
Organic compound						
HMDB0304531	Vanillylmandelate	1202	8816	1.12 (1.06–1.19)	1.2 × 10^−4^	No
Xenobiotics						
-	5-hydroxymethyl-2-furoylcarnitine	953	7071	1.22 (1.14–1.30)	1.6 × 10^−9^	Yes
-	2-hydroxyfluorene sulfate	932	6556	1.30 (1.22–1.38)	8.5 × 10^−16^	Yes

## Data Availability

The data that support the findings of this study are available from the corresponding authors [M.L. and L.F.S.] upon reasonable request.

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
