# Peer review of "Metabolomics-Based Machine Learning for Predicting Mortality: Unveiling Multisystem Impacts on Health"

_ijms, 2024, doi:10.3390/ijms252111636_

Round 1

Reviewer 1 Report

Comments and Suggestions for Authors

The work studied the effect of metabolomics on all-cause mortality, reporting several novel metabolites associated with an increased risk of mortality. Although the studied problem is interesting, I have the following concerns that need to be addressed. Some of major comments:

 1. Full terms of the abbreviation should be written out when it is first used, e.g., METSIM, SHAP, etc.

2. The writing of numbers is incorrect, "10.197", "13.512", "8.851", "1.346", etc.

3. The current Introduction only introduces the research background, please include and clearly state the current research status, existing challenges, innovations and contributions of proposed study.

4. It is recommended to indicate the ROC-AUC value in the legend of Figure 1 to make it clearer to the reader.

5. There is a problem with extra spaces, e.g., lines 124, 351, 358, etc.

6. It is recommended that Sections 2.4 and 2.5 could be combined into one section.

7. You say "945 metabolites" in line 370, but you say "944 metabolites" in Figure S1. Please be rigorous.

8. Please conduct a thorough proofreading to improve readability and professionalism. Please ensure that there are no logical and unnecessary errors.

9. In the Conclusions section, you must systematically: (1) highlight the theoretical and practical implications of your research; (2) discuss research contributions; (3) indicate practical advantages; (4) discuss research limitations; and (5) supply 2-3 solid and insightful future research suggestions.

10. The format of references is not uniform.

11. The DOI of reference 12 is incorrect, please pay attention to this problem in other references.

Author Response

Reviewer 1

Open Review

Quality of English Language

(x) The quality of English does not limit my understanding of the research.
( ) The English could be improved to more clearly express the research.

Yes

Can be improved

Must be improved

Not applicable

Does the introduction provide sufficient background and include all relevant references?

( )

(x)

( )

( )

Is the research design appropriate?

( )

(x)

( )

( )

Are the methods adequately described?

(x)

( )

( )

( )

Are the results clearly presented?

(x)

( )

( )

( )

Are the conclusions supported by the results?

(x)

( )

( )

( )

Comments and Suggestions for Authors

The work studied the effect of metabolomics on all-cause mortality, reporting several novel metabolites associated with an increased risk of mortality. Although the studied problem is interesting, I have the following concerns that need to be addressed. Some of major comments:

  1. Full terms of the abbreviation should be written out when it is first used, e.g., METSIM, SHAP, etc.

Reply: We have now corrected the full terms of abbreviations

  1. The writing of numbers is incorrect, "10.197", "13.512", "8.851", "1.346", etc.

Reply: We have now revised the writing of numbers

  1. The current Introduction only introduces the research background, please include and clearly state the current research status, existing challenges, innovations and contributions of proposed study.

Reply: We have clarified the current research status, existing challenges, innovations and contributions of proposed study.

  1. It is recommended to indicate the ROC-AUC value in the legend of Figure 1 to make it clearer to

the reader.

Reply: We have added the ROC-AUC value in the legend of Figure 1.

  1. There is a problem with extra spaces, e.g., lines 124, 351, 358, etc.

Reply: We have corrected the spaces including lines 124, 351, 358, etc.

  1. It is recommended that Sections 2.4 and 2.5 could be combined into one section.

Reply: We have combined Sections 2.4 and 2.5 into one section

  1. You say "945 metabolites" in line 370, but you say "944 metabolites" in Figure S1.

Please be rigorous.

Reply: We have corrected the number of metabolites in Figure S1 which is 945.

  1. Please conduct a thorough proofreading to improve readability and professionalism.

Please   ensure that there are no logical and unnecessary errors.

Reply: We have focused more on proofreading to improve readability and professionalism.

  1. In the Conclusions section, you must systematically: (1) highlight the theoretical and practical implications of your research; (2) discuss research contributions; (3) indicate practical advantages; (4) discuss research limitations; and (5) supply 2-3 solid and insightful future research suggestions.

Reply: We have rewritten the Conclusion section according to the advice of the Reviewer.

  1. The format of references is not uniform.

Reply: We have formatted references according to the style of the Journal.

  1. The DOI of reference 12 is incorrect, please pay attention to this problem in other references.

Reply: We have corrected the reference 12 and other references.

Submission Date

03 October 2024

Date of this review

09 Oct 2024 05:57:42

Reviewer 2 Report

Comments and Suggestions for Authors

Review of the Paper: "Metabolomics-Based Machine Learning for Predicting Mortality: Unveiling Multisystem Impacts on Health"

Strengths:

  1. The manuscript tackles an important clinical issue: predicting long-term mortality using a combination of metabolomics and machine learning (ML). The use of over 1,000 metabolites from a large cohort (METSIM) combined with multiple ML models (Logistic Regression, XGBoost, SVM) provides a well-rounded approach.
  2. This comprehensive dataset and method make the study particularly valuable for its breadth and potential clinical utility.
  3. The integration of conventional Cox regression alongside modern ML techniques helps validate the findings, and the cross-validation of ML models offers robustness to the statistical models used.

Recommendations:

  1. The discussion on how these findings translate into clinical practice is lacking. For instance, can the metabolites you identified be used to develop a risk score that improves mortality prediction compared to traditional clinical measures?
  2. Expanding on the potential clinical implications and future applicability (e.g., in cardiovascular disease management) would give more depth to the discussion.
  3. The manuscript provides a detailed description of the ML methods but lacks clarity on how the feature selection was precisely conducted for each model. For example, how were the top 32 metabolites finally chosen?
  4. Were thresholds set for feature importance across different models, or were they combined manually?
  5. A deeper exploration of how the dysregulation of these pathways contributes to mortality would strengthen the biological narrative. For instance, how do these metabolites directly affect known mechanisms of cardiovascular or renal failure?
  6. Including a graphical representation of the metabolic pathways affected by the identified metabolites, or heatmaps showing correlations between specific metabolites and mortality, would add clarity for readers unfamiliar with the underlying biology.
  7. The manuscript states that hyperparameter tuning was used to prevent overfitting, but how overfitting was evaluated? Was there a regularization technique applied in the ML models, particularly in XGBoost and SVM?
  8. Do short-term predictors reflect acute processes, while long-term predictors reflect chronic conditions?
  9. Were there specific clinical covariates (e.g., age, BMI) that interacted significantly with the identified metabolites?

Author Response

Reviewer 2

Open Review

Quality of English Language

(x) The quality of English does not limit my understanding of the research.
( ) The English could be improved to more clearly express the research.

Yes

Can be improved

Must be improved

Not applicable

Does the introduction provide sufficient background and include all relevant references?

( )

(x)

( )

( )

Is the research design appropriate?

( )

(x)

( )

( )

Are the methods adequately described?

( )

(x)

( )

( )

Are the results clearly presented?

( )

(x)

( )

( )

Are the conclusions supported by the results?

( )

(x)

( )

( )

Comments and Suggestions for Authors

Review of the Paper: "Metabolomics-Based Machine Learning for Predicting Mortality: Unveiling Multisystem Impacts on Health"

Strengths:

  1. The manuscript tackles an important clinical issue: predicting long-term mortality using a combination of metabolomics and machine learning (ML). The use of over 1,000 metabolites from a large cohort (METSIM) combined with multiple ML models (Logistic Regression, XGBoost, SVM) provides a well-rounded approach.
  2. This comprehensive dataset and method make the study particularly valuable for its breadth and potential clinical utility.
  3. The integration of conventional Cox regression alongside modern ML techniques helps validate the findings, and the cross-validation of ML models offers robustness to the statistical models used.

Replay: We thank the Reviewer for these positive comments.

Reviewer 2

Recommendations:

  1. The discussion on how these findings translate into clinical practice is lacking. For instance, can the metabolites you identified be used to develop a risk score that improves mortality prediction compared to traditional clinical measures?

Reply: We thank the Reviewer foe this comment. In fact, our Cox regression gives the answer to the Reviewer’s question. On page 6 (lines 169-177) we present the results of Cox regression analysis as follows: Clinical and laboratory measurements including age, BMI, waist circumference, smoking, systolic blood pressure, LDLC, total triglycerides, fasting glucose, hs-CRP, creatinine, T2D and UAE alone significantly increased the risk of mortality (HR 1.76, 1.60-1.94, p=1.7E-29). The 25 metabolites alone increased the risk of mortality also significantly (HR 1.89, 1.68-2.12, p= 5.2E-27). In the model including both clinical and laboratory measurements and 25 metabolites the risk of mortality further increased (HR 2.00, 1.81-2.22, p=3.4E-42) suggesting that the 25 metabolites increased the risk of mortality beyond the clinical and laboratory risk factors for mortality.  

This analysis was done to use Cox regression as a prediction model for the risk of mortality. We compared the prediction of clinical factors, metabolites, and a combination of these two as a prediction of mortality which could be used also in clinical practice. Clinical factors alone increased the risk of mortality by 76% (HR 1.76), 25 metabolites alone increased the risk of mortality by 89% (HR 1.89), and together by 200 % (HR 2.00). In these analyses metabolites were included in the model using their beta from the linear regression.

Consequently, our findings could be used also in clinical work.  We have now added this in Discussion.

  1. Expanding on the potential clinical implications and future applicability (e.g., in cardiovascular disease management) would give more depth to the discussion.

Reply: Our study showed for the first time that the metabolites identified were especially regulating the risk of cardiovascular diseases. In Discussion we give a more detail clinical implications of our findings.

  1. The manuscript provides a detailed description of the ML methods but lacks clarity on how the feature selection was precisely conducted for each model. For example, how were the top 32 metabolites finally chosen?

Reply: The selection of the 32 metabolites was performed by applying a multi-step feature selection process using the three distinct methods, logistic regression, Welch's t-test, and XGBoost. Each method was used to rank the metabolites independently. The top metabolites were selected based on the lowest q-values from the Welch's t-test and the highest ROC-AUC values from logistic regression for each individual feature (metabolite). XGBoost was used to rank the metabolites by importance values, selecting those with a SHAP feature importance value greater than 0.012, resulting in 154 metabolites.

The final set of the most impactful 32 metabolites was selected from the intersection of the top-ranked metabolites identified by these three methods. In other words, logistic regression, Welch's t-test, and XGBoost selected the identical set of 32 metabolites which gives strong evidence that our calculations are reliable.

  1. Were thresholds set for feature importance across different models, or were they combined manually?

Reply: As explained in our response in our response to criticisms in 3, three methods were employed (logistic regression, XGBoost, the Welch's t-test) in the feature selection process. A specific threshold for feature importance, SHAP feature importance 0.012, was applied in the XGBoost model, which was trained on a dataset of 945 metabolites. We used only one feature importance threshold for XGBoost model. For logistic regression, the threshold for metabolite selection was based on the ROC-AUC value to assess the discrimination ability of each metabolite. The Welch's t-test was performed for each metabolite individually and its threshold stated for statistical significance.

  1. A deeper exploration of how the dysregulation of these pathways contributes to mortality would strengthen the biological narrative. For instance, how do these metabolites directly affect known mechanisms of cardiovascular or renal failure?

Reply: Our discussion describes in detail the current knowledge about the metabolites associated with the risk of different diseases, including cardiovascular and renal diseases. There is no information available about the exact causal mechanisms. Fig. S8 (Figures A-C) shows the main metabolic pathways for the metabolites associated with increased and decreased risk of mortality.

  1. Including a graphical representation of the metabolic pathways affected by the identified metabolites, or heatmaps showing correlations between specific metabolites and mortality, would add clarity for readers unfamiliar with the underlying biology.

Reply: Fig. 3 summarizes the current knowledge. We should remember that our study, as well as all previously published studies about the metabolite-disease association, do not show causality. However, these studies are important because they link metabolites with previously unknown diseases. Furthermore, we do not know all the pathways the metabolites are belonging to. Supplemental Figure (Fig. S10) and Fig. 3. show the most important known metabolic pathways of the metabolites associated with an increased or decreased risk of mortality.

  1. The manuscript states that hyperparameter tuning was used to prevent overfitting, but how overfitting was evaluated? Was there a regularization technique applied in the ML models, particularly in XGBoost and SVM?

Reply:  Overfitting was assessed by comparing the performance metrics between the training set and the independent test set in XGBoost. No significant drop in the performance of the test set was found which indicates good generalization. Additionally, overfitting was evaluated using sub-splits of the independent test set (3%, 5%, and 7%), and consistent ROC-AUC values across these sub-splits confirmed the models' robustness and generalizability.

In XGBoost, several regularization techniques were applied to control overfitting, including L2 regularization (reg_lambda=1), subsampling (subsample=0.457), feature subsampling (colsample_bytree=0.52), minimum child weight (min_child_weight=2), and limiting tree depth (max_depth=5). For logistic regression, L2 regularization with a moderate strength (C=1.0) was used, and models were built for each metabolite separately, reducing complexity and minimizing the need for strong regularization. The class imbalance, which can lead to overfitting, was also addressed to further improve model performance. Similarly, in the SVM model, L2 regularization (C=1.0) was applied, with class balancing to reduce the risk of overfitting.

  1. Do short-term predictors reflect acute processes, while long-term predictors reflect chronic conditions?

        Reply: Yes.

  1. Were there specific clinical covariates (e.g., age, BMI) that interacted significantly with the identified metabolites?

              Reply: We did not find covariates that interacted significantly with the identified metabolites.

Submission Date

03 October 2024

Date of this review

11 Oct 2024 06:52:22

Round 2

Reviewer 1 Report

Comments and Suggestions for Authors

By revising, you have corrected most of the comments with your paper, but there are still too many basic mistakes.

1. The writing of some numbers still be incorrect, such as "n=8.851" in line 89.

2. In column p of Tables 1 and 2, it should be the mathematical notation "×" instead of the letter "x". The same problem also appears in lines 178, 179, 181.

3. The mathematical notation "×" is missing from "<1.0 10-20" in row 4 of Table 2. Additionally, you wrote "Nov" in Table 2, but you should have written "Novel". You didn't write the whole word.

3. Please delete the annotations in the manuscript.

4. You still say "944 metabolites" in Figure S1. You did not correct the error.

5. Reference 12 has an incorrect DOI. You did not revise it to the correct DOI, but instead deleted it. Additionally, the DOIs for other references were also removed. I am not satisfied with your response.

Author Response

Open Review

Quality of English Language

(x) The quality of English does not limit my understanding of the research.
( ) The English could be improved to more clearly express the research.

Yes

Can be improved

Must be

improved

Not applicable

Does the introduction provide sufficient background and include all relevant references?

( )

(x)

( )

( )

Is the research design appropriate?

( )

(x)

( )

( )

Are the methods adequately described?

(x)

( )

( )

( )

Are the results clearly presented?

(x)

( )

( )

( )

Are the conclusions supported by the results?

(x)

( )

( )

( )

Comments and Suggestions for Authors

By revising, you have corrected most of the comments with your paper, but there are still too many basic mistakes.

  1. The writing of some numbers still be incorrect, such as "n=8.851" in line 89.

Answer: We corrected the number for 8851 in line 89.

  1. In column pof Tables 1 and 2, it should be the mathematical notation "×" instead of the letter "x". The same problem also appears in lines 178, 179, 181.

Answer: We replaced “x” with “×”.

  1. The mathematical notation "×" is missing from "<1.0 10-20" in row 4 of Table 2. Additionally, you wrote "Nov" in Table 2, but you should have written "Novel". You didn't write the whole word.

Answer: We include the “x” missing in the row 4. We also included “Novel” in the Table 2.

  1. Please delete the annotations in the manuscript.

Answer: We have now deleted the annotations.

  1. You still say "944 metabolites" in Figure S1. You did not correct the error.

Answer: We replaced the Figure S1 (945 metabolites).

  1. Reference 12 has an incorrect DOI. You did not revise it to the correct DOI, but instead deleted it. Additionally, the DOIs for other references were also removed. I am not satisfied with your response.

We followed the recommendation for the International Journal of Molecular Sciences guidelines, as found in the website:

 “Your references may be in any style, provided that you use the consistent formatting throughout. It is essential to include author(s) name(s), journal or book title, article or chapter title (where required), year of publication, volume and issue (where appropriate) and pagination. DOI numbers (Digital Object Identifier) are not mandatory but highly encouraged.” 

https://www.mdpi.com/journal/ijms/instructions

Submission Date

03 October 2024

Date of this review

21 Oct 2024 11:50:53
